# Cross-Cultural Adaptation and Lingual Validation of the Beliefs about Medicines Questionnaire (BMQ)-Specific for Cholesterol Lowering Drugs in the Visegrad Countries

**DOI:** 10.3390/ijerph17207616

**Published:** 2020-10-19

**Authors:** Klára Boruzs, Viktor Dombrádi, János Sándor, Gábor Bányai, Robert Horne, Klára Bíró, Attila Nagy

**Affiliations:** 1Department of Health Systems Management and Quality Management for Health Care, Faculty of Public Health, University of Debrecen, 4032 Debrecen, Hungary; boruzsklara@gmail.com (K.B.); dombradi.viktor@gmail.com (V.D.); banyai@med.unideb.hu (G.B.); kbiro@med.unideb.hu (K.B.); 2Department of Public Health and Epidemiology, Faculty of Medicine, University of Debrecen, 4028 Debrecen, Hungary; sandor.janos@sph.unideb.hu; 3Centre for Behavioural Medicine, UCL School of Pharmacy, University College London, London WC1E 6BT, UK; r.horne@ucl.ac.uk; 4Faculty of Public Health, University of Debrecen, 4028 Debrecen, Hungary

**Keywords:** cardiology, epidemiology, preventive medicine, vascular medicine

## Abstract

The goal of this study was to translate the Beliefs about Medicines Questionnaire—Specific (BMQ-Specific) for cholesterol-lowering drugs, into the Hungarian, Slovak, Czech and Polish languages and test their reliability with statistical methods. For this purpose, Cronbach’s alpha, confirmatory and exploratory factor analyses were conducted. The analyses included 235 Czech, 205 Hungarian, 200 Polish, and 200 Slovak respondents, all of whom were taking cholesterol-lowering drugs. The translations from English into the target languages were always done by two independent translators. As part of the validation process these translations were pilot tested and after the necessary alterations, they were translated back into English by a third translator. After the approval by the creator of the questionnaire, nationwide surveys were conducted in all four countries. The results of the confirmatory factor analysis were exceptionally good for the Czech and Slovak translations, while the Polish and Hungarian translations marginally crossed the predetermined thresholds. With the exception of a single Polish question, the results of the exploratory factor analysis were deemed acceptable. The translated versions of BMQ-Specific are reliable and valid tools to assess patients’ beliefs about medication, especially medication adherence among patients taking cholesterol-lowering medication. A comparison between the four countries with this questionnaire is now possible.

## 1. Introduction

The health status of the population of Central and Eastern European countries is worse than the health status of the populations of high-income member states within the European Union. Although epidemiological data strongly suggest that cardiovascular mortality is higher in Central and Eastern European countries [1], only one study has been conducted in these countries thus far [2].

In a systematic review it was found that several factors may contribute to inadequate statin therapy. One factor is that doctors do not prescribe statin or use statin treatment with a lower intensity than the dose specified in the guidelines because they are afraid of the harmful side effects of the drug. Another influencing factor may be the fear of patients related to statin therapy because information from communication channels often suggests that a number of serious adverse side effects may occur when taking this kind of medication. Therefore, some patients stop taking statins after a while or do not follow the dose prescribed by their doctor, resulting in worse adherence. According to a survey conducted in the United States, 62% of patients stopped taking statins because they thought statins had side effects, and one-third of these patients stopped taking the drug without even consulting their doctor [3].

In patients without established cardiovascular disease (CVD) but with cardiovascular risk factors, statin use was associated with significantly improved survival rate and large reductions in the risk of major cardiovascular events [4]. It was demonstrated that every 1 mmol/L decrease in LDL cholesterol results in a 21% decrease in cardiovascular events [5]. Therefore, statins are considered the first choice of medication for patients with hypercholesterolemia or combined hyperlipidemia, to reduce their risk of CVDs [6]. Due to improper use of statin medication, the desired health benefits cannot be achieved. One of the main reasons for not taking the medicine is the aforementioned attitude towards the drug. If we can measure this with a reliable questionnaire and utilize it in multiple countries, then we will be able to identify good policies and apply them in other countries as well.

One such reliable questionnaire used in international research studies is the Beliefs about Medicines Questionnaire (BMQ) [7]. This instrument was introduced by Horne et al. in 1999 and was originally developed in English. It comprises two brief questionnaires: BMQ-Specific and BMQ-General. The BMQ-Specific was designed to assess key beliefs about influencing engagement with prescribed medicines. It comprises two scales: Specific-Necessity and Specific-Concerns. Specific-Necessity assesses patients’ beliefs about their personal need for the prescribed medicine and how important they perceive it to be, while the Specific-Concerns assesses concerns about the treatment. The BMQ-General assesses social representations about pharmaceuticals as a class of treatment. This originally comprised two scales assessing beliefs about the degree to which medicines are fundamentally harmful (General-Harm scale), and addictive substances that are prescribed too often (General-Overuse scale). A General-Benefit scale assessing perceptions of the benefits of medicines was added later. The BMQ is used worldwide and has been translated and validated in several languages [8,9].

Studies have demonstrated that there is a relationship between BMQ scores and adherence behavior regarding medicine for various chronic diseases [7], including asthma [10], chronic obstructive pulmonary disease (COPD) [11], psoriasis [12], hypertension [13] and rheumatoid arthritis treatment [14]. Although a survey was conducted on cholesterol-lowering treatment, only the General version of the BMQ was used in that study [15].

Therefore, the primary aim of this study was to conduct a cultural adaptation and validation of the BMQ-Specific and test the reliability of the translation via statistical methods. These translations then can be applied in studies examining patients’ beliefs about cholesterol-lowering medicines in Hungary, Slovakia, the Czech Republic and Poland.

It is worth noting that validated translations already exist for the BMQ-Specific in both the Czech and Polish languages. The Czech version tested general drug use of patients with either diabetes, hypertension or rheumatic disease [16], and was later used to study the adherence to oral bisphosphonates [17]. In the Czech validation study, principal component analysis was performed to explore the structure of factors, and reliability was also investigated by Cronbach’s alpha [17]. The Polish version was first used in an international research study to better understand the adherence to psychotropic medications during pregnancy [18]. Later, the Polish translation was also adapted and validated for cardiovascular patients and medical students, and its reliability was tested using similar methods, such as Cronbach’s alpha and Pearson correlation besides the confirmatory factor analysis in the aforementioned study [19]. Thus, in these two languages the current study should be seen as a supplement regarding cholesterol-lowering medication.

## 2. Materials and Methods

### 2.1. Data Collection and Study Settings

Neither patients nor the public were involved in the design, conduct, reporting or dissemination plans of our research. In all four countries the data gathering method was identical. An online questionnaire was sent out to the citizens by private polling companies. These citizens were part of an online market research panel that is used by the companies to conduct surveys at an international level [20]. The panels were made to represent the population of the particular country and preliminary consent was needed in order to be a member. The panel members were gradually invited to take part in this study until 1000 citizens had completed the questionnaire in each country. Representativeness was taken into consideration in terms of age (18 and older), gender, size of settlement and region. This was achieved by an automatic data collection information technology system. For example, if too many males started to complete the questionnaire, then the system recognized the over representativeness and intervened by randomly sending the questionnaires to primarily female panel members. Because not all of the participants had been taking cholesterol-lowering drugs, a smaller sample was randomly selected—taking representativity into account—of which all participants answered that they were taking such medicine. Thus, the current study consists of 205 Hungarian, 200 Slovak, 235 Czech and 200 Polish responses, representative of the previously described factors.

Ethics approval: All respondents of this study were approached via online market research panels by the companies SZLEM Service L. *p*. and Český Národní Panel Ltd. and were fully anonymous to all of the authors of the study. Written informed consent for the study was obtained from all respondents prior to the study by SZLEM Service L. *p*. and Český Národní Panel Ltd. The study was approved by the Scientific Research and Ethics Committee of the Medical Research Council in Hungary (ETT TUKEB 55704–5/2017/EKU) and by the Ethics Committee of the Czech University Hospital Hradec Kralove in Czechia (Ref. number: 201802 S15 P). For the Slovak ethical approval, the Ethics Committee of the Ministry of Health and the Ethics Committee of the University of Kosice were contacted. Both committees stated that non-interventional studies and market research studies do not require ethics approval. Regarding the Polish ethical approval, the Bioethics Committee of the Ministry of Health was contacted, which also stated that non-interventional studies and market research studies do not require ethics approval.

### 2.2. Translation and Language Adaptation

A common process was used to translate and adapt the BMQ-Specific across the four languages. The original English version [7] was translated by two independent translators into their native language. Once completed, the two translations were merged into a single translation and were altered in a way that the questions were directed to cholesterol-lowering drugs. Afterwards, as part of the validation process, each translation was pilot tested by ten native-speaking citizens and the translations were revised based on their feedback. A third independent translator translated the questionnaire back to English. Once all four questionnaires were translated back to the original language, they were sent to the creator of the BMQ, Robert Horne, to assess if the back translations had the same meaning as the original. In all four cases the back translations were deemed adequate.

### 2.3. Statistical Analysis

The four translations were treated independently from one another in all statistical analyses. Besides the general descriptive statistics, Cronbach’s alpha was calculated to investigate the internal reliability of both the Necessity and Concerns scales. The internal reliability was deemed acceptable if the alpha value was equal to or above 0.70 [21]. Chi-square (X2), ratio of Chi-square to degrees of freedom (X2/df), Comparative Fit Index (CFI), Tucker-Lewis Index (TLI), Standardized Root Mean Square Residual (SRMR) and Root Mean Square Error of Approximation (RMSEA) were used to conduct the confirmatory factor analysis. Based on the literature, the recommended value of the X2/df ratio should be less than 3 [22], the CFI and the TLI values should be above 0.90 [23], the SRMR should be less than 0.08, and for the RMSEA, if the value is less than 0.08 [24], then it is considered acceptable, while a value less than 0.10 is only marginally acceptable [25]. When assessing the factor loadings, 0.4 was considered the minimum acceptable value [26]. Intercooled Stata version 13.0 was used for the statistical analyses.

## 3. Results

The patients’ demographic characteristics are presented in Table 1. In Hungary, Slovakia and the Czech Republic most participants were female (Hungary: 61%, Slovakia: 55.5% and the Czech Republic: 59.1%), while in Poland most of the participants were male (53%). Furthermore, in all four countries the majority of the participants were in the 55–65 age group (Hungary: 59%, Slovakia: 40.5%, Czech Republic: 61.7% and Poland: 38%). Finally, while more than one-third of Hungarian, Slovak and Polish (Hungary: 34.6%, Slovakia: 37% and Poland: 34%) respondents had a college or university degree, only 17% of Czech respondents reported having such an education degree.

In Table 2 the descriptive analysis of the BMQ-Specific items is shown. The Cronbach’s alpha values were between 0.782 and 0.851 (Table 3). Thus, the scales of none of the translations went below the predetermined 0.7 threshold.

The confirmatory factor analysis (Table 4) revealed that the model fit was satisfactory for the Czech (CFI = 0.919; TLI = 0.896; SRMR = 0.057; RMSEA = 0.082) and Slovak (CFI = 0.926; TLI = 0.905; SRMR = 0.063; RMSEA = 0.084) translations, while the Polish (CFI = 0.863; TLI = 0.825; SRMR = 0.082; RMSEA = 0.123) and Hungarian (CFI = 0.866; TLI = 0.829; SRMR = 0.076; RMSEA = 0.108) translations marginally crossed the predetermined thresholds.

The results of the exploratory factor analysis of the BMQ-Specific are reported in Table 5. Acceptable level of consistency was shown between the items in the four countries except for the Polish translation of the Concerns scale, as the value of one question went below the acceptable threshold of 0.4 (0.257 for item 2).

## 4. Discussion

### 4.1. Main Findings

Validation of a questionnaire and testing its reliability are complex and comprise multiple methods [27]. As indicated at the beginning of the paper, the primary aim of this study was to present doctors and researchers from Hungary, Slovakia, the Czech Republic and Poland with a questionnaire that can measure the attitude towards cholesterol-lowering medication. According to our findings, all four translations of the BMQ-Specific for cholesterol-lowering drugs can be used as a reliable tool for this purpose. In addition, this is the first study to describe the simultaneous cross-cultural validation and adaptation of the Hungarian, Slovak, Czech and Polish versions of BMQ-Specific for cholesterol-lowering medication.

The original BMQ study by Horne et al. [7] suggested that the two-factor solutions can explain 51% of the variances. Furthermore, Cronbach’s alpha values obtained for Necessity and Concerns were 0.86 and 0.65, respectively. In a reliability analysis of a questionnaire, it is considered satisfactory when the Cronbach’s alpha coefficient is greater than 0.7, although some studies suggest that a value above 0.6 is also sufficient [28,29]. Previous research evaluating the translated versions of the BMQ had demonstrated similar values for internal consistency, with the Portuguese version and the Sinhalese version having a Cronbach’s alpha coefficient of 0.66 and 0.65, respectively [30,31]. Although the Cronbach’s alpha value for the Czech translation in our analysis was somewhat lower than in the other Czech study (0.85 for Necessity and 0.82 for Concerns), the high values indicate that both versions are suitable for measurement [17]. In the Polish version the alpha coefficient for Necessity was between 0.79 and 0.82, while for Concerns it was between 0.65 and 0.70, depending on the group participating in the study [20]. Thus, the alpha coefficient was higher for the Polish translation in our study. Overall, the translated questionnaires demonstrated good reliability, temporal stability and validity.

Considering the confirmatory factor analysis, a similar French study revealed CFI as 0.89 and RMSEA as 0.08 [32]. Furthermore, a Persian study revealed CFI as 0.96 and RMSEA as 0.07 [14]. In our own study, the values for the CFI, TLI, SRMR and RMSEA for the Czech and Slovak translations were all within the predetermined thresholds. However, these values for both the Hungarian and Polish versions did not meet these criteria. Although these can be considered only marginal deviations, future studies using these translations should report this as a limiting factor. Since many of the Czech or the Polish studies that included the BMQ conducted confirmatory factor analysis, a comparison with our own study was not possible [17,18,33]. Nevertheless, comparison with the recently published Polish BMQ validation for inpatient and outpatient cardiovascular patients, as well as medical students, is possible [20]. In this study, when the authors used a four-factor model, the model fit was satisfactory for the inpatient group (CFI = 0.894; TLI = 0.871; SRMR = 0.079; RMSEA = 0.055) and for the outpatient group it did not meet the conventional fit criteria (CFI = 0.850; TLI = 0.819; SRMR = 0.090; RMSEA = 0.067), while for the medical students it was far from the conventional fit criteria (CFI = 0.799; TLI = 0.759; SRMR = 0.093; RMSEA = 0.082).

Finally, in our study, with the exception of a single Polish question the factor loadings were all above the acceptable threshold. However, for the sake of uniformity, the aforementioned question was not altered.

### 4.2. Strengths and Limitations

The translation and testing of the BMQ-Specific for cholesterol-lowering drugs was done in four languages simultaneously. International standards were used by testing their reliability with various statistical methods. While the Slovak and Czech translations met the suggested thresholds for the confirmatory factor statistics, the Hungarian and Polish translations marginally crossed the predetermined thresholds. The factor loading of a single Polish question was below the acceptable threshold.

## 5. Conclusions

Based on the results of our statistical analyses the Hungarian, Slovak, Czech and Polish versions of BMQ-Specific are all reliable, and based on the four pilot tests, these are also valid tools to assess patients’ beliefs about medication adherence in relation to cholesterol-lowering medication. A comparison between the four countries is now possible.

## Figures and Tables

**Table 1 ijerph-17-07616-t001:** Demographic characteristics of the study sample.

	Hungary		Slovakia		Czech Republic		Poland	
	n	%	n	%	n	%	n	%
**Gender**								
Female	125	61.0%	111	55.5%	139	59.1%	94	47.0%
Male	80	39.0%	89	44.5%	96	40.9%	106	53.0%
**Age**								
18–24	0	0.0%	2	1.0%	1	0.4%	2	1.0%
25–34	1	0.5%	6	3.0%	8	3.4%	11	5.5%
35–44	10	4.9%	28	14.0%	15	6.4%	42	21.0%
45–54	29	14.1%	58	29.0%	55	23.4%	55	27.5%
55–65	121	59.0%	81	40.5%	145	61.7%	76	38.0%
+65	44	21.5%	25	12.5%	11	4.7%	14	7.0%
**Education**								
Primary school	30	14.6%	24	12.0%	63	26.8%	6	3.0%
High school	104	50.7%	102	51.0%	132	56.2%	126	63.0%
College or university	71	34.6%	74	37.0%	40	17.0%	68	34.0%

**Table 2 ijerph-17-07616-t002:** Descriptive statistics of the Beliefs about Medicines Questionnaire (BMQ)-Specific items.

Item	Hungary		Slovakia		Czech Republic		Poland	
	Mean	SD	Mean	SD	Mean	SD	Mean	SD
**Necessity**								
1	2.67	1.14	3.12	1.11	2.86	1.21	3.37	1.25
3	2.37	1.26	2.56	1.17	2.21	1.16	2.83	1.28
4	2.38	1.22	2.61	1.11	2.49	1.14	2.88	1.33
7	2.51	1.17	2.91	1.12	2.94	1.11	3.09	1.27
10	2.88	1.17	3.04	1.20	3.57	1.01	3.26	1.19
**Concerns**								
2	3.14	1.35	3.03	1.27	3.33	1.24	3.57	1.26
5	3.25	1.37	3.33	1.30	3.03	1.37	3.27	1.32
6	2.91	1.33	2.84	1.16	2.75	1.19	2.49	1.23
8	2.12	1.21	2.24	1.13	2.08	1.17	2.18	1.19
9	2.41	1.37	2.44	1.34	2.43	1.26	2.56	1.29
11	2.41	1.32	2.17	1.26	1.90	1.10	2.42	1.20

**Table 3 ijerph-17-07616-t003:** Internal reliability of the two scales.

Country	Scale	Cronbach’s Alpha
**Hungary**	Necessity	0.845
	Concerns	0.782
**Slovakia**	Necessity	0.851
	Concerns	0.818
**Czech Republic**	Necessity	0.794
	Concerns	0.817
**Poland**	Necessity	0.871
	Concerns	0.789

**Table 4 ijerph-17-07616-t004:** Fit statistics for BMQ-Specific.

Fit Measure	Two-Factor (11 Items)			
	Hungary	Slovakia	Czech Republic	Poland
x^2^	146.58	104.13	110.28	172.27
*p*-Value	*p* < 0.001	*p* < 0.001	*p* < 0.001	*p* < 0.001
df	43	43	43	43
x^2^/df	3.41	2.42	2.56	4.01
CFI	0.866	0.926	0.919	0.863
TLI	0.829	0.905	0.896	0.825
SRMR	0.076	0.063	0.057	0.082
RMSEA (90% CI)	0.108 (0.089–0.128)	0.084 (0.064–0.105)	0.082 (0.063–0.101)	0.123 (0.104–0.142)
*p*-Value (RMSEA < 0.05)	*p* < 0.001	0.004	0.004	*p* < 0.001

CFI = Comparative Fit Index; TLI = Tucker-Lewis Index; SRMR = Standardized Root Mean Square Residual; RMSEA = Root Mean Square Error of Approximation; CI = Confidence Interval.

**Table 5 ijerph-17-07616-t005:** Exploratory factor analysis of the BMQ-Specific items.

Item	Hungary		Slovakia		Czech Republic		Poland	
	Factor 1	Factor 2	Factor 1	Factor 2	Factor 1	Factor 2	Factor 1	Factor 2
**1**	0.667	0.125	0.671	0.045	0.628	0.131	0.756	0.027
**3**	0.715	0.206	0.722	0.188	0.699	0.182	0.765	0.318
**4**	0.774	0.246	0.770	0.243	0.771	0.197	0.761	0.238
**7**	0.733	0.131	0.801	0.091	0.670	0.189	0.723	0.188
**10**	0.690	−0.133	0.649	0.027	0.473	−0.052	0.680	0.081
**2**	0.194	0.581	0.186	0.596	0.116	0.594	0.441	0.257
**5**	0.076	0.566	0.221	0.578	0.192	0.613	0.390	0.562
**6**	0.031	0.637	−0.036	0.577	0.090	0.532	0.209	0.527
**8**	0.087	0.761	0.050	0.827	0.035	0.840	0.027	0.771
**9**	0.227	0.498	0.168	0.689	0.288	0.649	0.322	0.692
**11**	−0.007	0.605	0.064	0.627	0.012	0.640	0.049	0.652

Note: The grey background indicates the scales in which a specific item can be found.

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
