# Peer review of "Cross-Cultural Adaptation and Lingual Validation of the Beliefs about Medicines Questionnaire (BMQ)-Specific for Cholesterol Lowering Drugs in the Visegrad Countries"

_ijerph, 2020, doi:10.3390/ijerph17207616_

Round 1
Reviewer 1 Report
A well written paper - but then it is biased in the sense that Rob Horne who developed the original questionnaire is also one of the authors. Quite a lot of stats in here- I am not an expert in statistics but it does make sense to me and also limitations of some results are mentioned.
Author Response
Dear Reviewer,
We would like to thank for taking your time and for providing useful advice on improving the content of the manuscript. As required, we are sending the ‘change tracked’ version of the revised manuscript.
Reviewer: A well written paper - but then it is biased in the sense that Rob Horne who developed the original questionnaire is also one of the authors. Quite a lot of stats in here- I am not an expert in statistics but it does make sense to me and also limitations of some results are mentioned.
We understand the concern you have brought up. Professor Robert Horne is a co-author of this paper in order to ensure that the translation and validation process of the BMQ is adequate. This is necessary to avoid poor translations of the BMQ to be available for researchers who might want to use this instrument in their own language. Also, Professor Robert Horne did not participate in the statistical analysis and thus did not influence the results.
Sincerely yours,
Attila Nagy on behalf of the authors
Reviewer 2 Report
This is a well carried out and well written up study, addressing a clear gap in the literature.
I have one suggestion for improvement. In the discussion section it may be helpful to ensure that you have distinguished clearly between reliability and validity. I don't think the statistical tests that you carried out would test for validity, only reliability. However, the processes that you used during translation would be a test of face and content validity. It would be good to make this distinction clearer in the text so that the readers can clearly see where you have you reached your conclusions.
Author Response
Dear Reviewer,
We would like to thank for taking your time and for providing useful advice on improving the content of the manuscript. As required, we are sending the ‘change tracked’ version of the revised manuscript.
Reviewer: This is a well carried out and well written up study, addressing a clear gap in the literature. I have one suggestion for improvement. In the discussion section it may be helpful to ensure that you have distinguished clearly between reliability and validity. I don't think the statistical tests that you carried out would test for validity, only reliability. However, the processes that you used during translation would be a test of face and content validity. It would be good to make this distinction clearer in the text so that the readers can clearly see where you have you reached your conclusions.
Response: Thank you for your positive remarks! Regarding the usage of terms ‘reliability’ and ‘validity’ we used them incorrectly in the original manuscript. We would like to apologize for that. To remedy this, thorough the manuscript we now use ‘reliability’ when referring to the statistical tests, and ‘validity’ when mentioning the pilot testing after the initial translations. The changes within the original manuscript are at lines 17-31 (abstract); 75-77; lines 84-86; lines 86-88; lines 110-112; line 158; lines 198-202; and lines 204-206.
Sincerely yours,
Attila Nagy on behalf of the authors
Reviewer 3 Report
Thank you for the submission. It does, however, need a small revision before being re-considered for possible publication in the Int. J. Environ. Res. Public Health. It is not very clear why the study was actually needed when the validation has previously been reported in very similar Czech patients (with hypertension and diabetes; many of whom would in fact take statins) (Matoulkova et al) and Polish cardiovascular patients (Karbownik). Would a large difference be expected with the other two countries?
The introduction is a little confusing in combining the prescribing of statins with patient adherence to statin therapy. The two are very different and should be more clearly differentiated.
The authors state that there have been no studies examining the BMQ and adherence with statin therapy. This is incorrect e.g. Bermingham M, Hayden J, Dawkins I, et al. Prospective analysis of LDL-C goal achievement and self-reported medication adherence among statin users in primary care. Clin Ther 2011;33:1180-9.
The English also needs careful revision.
Author Response
Dear Reviewer,
We would like to thank for taking your time and for providing useful advice on improving the content of the manuscript. As required, we are sending the ‘change tracked’ version of the revised manuscript.
Reviewer: Thank you for the submission. It does, however, need a small revision before being re-considered for possible publication in the Int. J. Environ. Res. Public Health.
Reviewer: It is not very clear why the study was actually needed when the validation has previously been reported in very similar Czech patients (with hypertension and diabetes; many of whom would in fact take statins) (Matoulkova et al)
Response 1: The study of Matoulkova et al. was disease specific, focusing on patients with diabetes, hypertension or rheumatic disease, while our study focused on statin medication. Your remark is correct that many of those who have hypertension and diabetes do take statins as well, however, they can take other medications also. Although there is some overlap, we believe that it would be beneficial to investigate the validity and reliability of the Czech translation of the BMQ-Specific focusing on cholesterol-lowering medications.
For clarification, in our revision we now mention that the study of Matoulkova et al. focused on the general usage of drugs only (original manuscript lines 79-80): “The Czech version tested general drug use of patients with either diabetes, hypertension or rheumatic disease [16], …”
Reviewer: Polish cardiovascular patients (Karbownik). Would a large difference be expected with the other two countries?
Response 2: At the 12th European Public Health Conference (EUPHA) we presented our initial findings as a poster comparing the answers of the four countries. The title of the poster was “Beliefs About Medicines: Differences in cholesterol treatment adherence among the Visegrad countries” and the abstract is available at:
https://academic.oup.com/eurpub/article/29/Supplement_4/ckz187.008/5623117 . According to our initial findings there were no differences observed regarding BMQ-Concerns, however, significant differences were found concerning BMQ-Necessity. However, these initial findings do not include confounders in the statistical analyses, thus, the results may change. A detailed analysis between the countries will be performed in the near future.
Reviewer: The introduction is a little confusing in combining the prescribing of statins with patient adherence to statin therapy. The two are very different and should be more clearly differentiated.
Response 3: We agree with your statement, thus, from lines 40-56 was have completely rephrased the paragraphs. We now mention, the inadequate statin therapy is due to two main factors: (1) the doctor does not prescribe the medicine in the first place, and (2) patients do no take the medicine due to various beliefs. The second factor creates the justification to investigate how patients related to statin medication.
Reviewer: The authors state that there have been no studies examining the BMQ and adherence with statin therapy. This is incorrect e.g. Bermingham M, Hayden J, Dawkins I, et al. Prospective analysis of LDL-C goal achievement and self-reported medication adherence among statin users in primary care. Clin Ther 2011;33:1180-9.
Response 4: Because Bermingham et al. only used BMQ-General – not BMQ-Specific – in their survey we did not mention their study. In retrospect, this was a mistake. At lines 73-74 we have deleted the sentence stating “However, no such studies have yet been conducted regarding cholesterol lowering treatment.” and instead wrote a new sentence with the proper citation: “Although a survey was conducted on cholesterol-lowering treatment, but only the General version of BMQ was used in that study [15].”
Reviewer: The English also needs careful revision.
Response 5: We have asked one of our colleagues who speaks fluently English to carefully read the manuscript and check for grammatical errors. All the necessary changes were made with ‘track changes’.
Sincerely yours,
Attila Nagy on behalf of the authors
Reviewer 4 Report
Thank you for this study. BMQ is a great scale and this is a great study.
Lines 73 - 74: BMQ has been used in statins. Please add reference(s). Or may be the authors meant to say it has not been used in Eastern European population. Please clarify.
Author Response
Dear Reviewer,
We would like to thank for taking your time and for providing useful advice on improving the content of the manuscript. As required, we are sending the ‘change tracked’ version of the revised manuscript.
Reviewer: Lines 73 - 74: BMQ has been used in statins. Please add reference(s). Or may be the authors meant to say it has not been used in Eastern European population. Please clarify.
Response: Your remark is correct. The BMQ was indeed used by Bermingham et al. regarding statins. However, they only used BMQ-General and for some reason, did not include BMQ-Specific in their study. Therefore, for clarification, we deleted the original statement at lines 73-74 and added a new one: “Although a survey was conducted on cholesterol-lowering treatment, but only the General version of BMQ was used in that study [15].”
Sincerely yours,
Attila Nagy on behalf of the authors
Round 2
Reviewer 3 Report
Thank you for the revision